# A Single Dose of Live-Attenuated Rift Valley Fever Virus Vector Expressing Peste Des Petits Ruminants Virus (PPRV) H or F Antigens Induces Immunity in Sheep

**DOI:** 10.3390/vaccines13101039

**Published:** 2025-10-09

**Authors:** Sandra Moreno, Gema Lorenzo, Verónica Martín, Celia Alonso, Friedemann Weber, Belén Borrego, Alejandro Brun

**Affiliations:** 1Centro de Investigación en Sanidad Animal (CISA), Instituto Nacional de Investigación y Tecnología Agraria y Alimentaria, Consejo Superior de Investigaciones Científicas (INIA-CSIC), Valdeolmos, 28130 Madrid, Spain; lorenzo.gema@inia.csic.es (G.L.); veronica.martin@inia.csic.es (V.M.); celia.alonso@inia.csic.es (C.A.); borrego@inia.csic.es (B.B.); 2Institute for Virology, FB10-Veterinary Medicine, Justus-Liebig University, 35392 Giessen, Germany; friedemann.weber@vetmed.uni-giessen.de

**Keywords:** viral vector bivalent vaccine, Rift Valley fever virus, peste des petits ruminant virus

## Abstract

Introduction/Background: Rift Valley fever virus (RVFV) and peste des petits ruminants virus (PPRV) are significant pathogens affecting small ruminants, causing substantial economic losses in the affected regions. The development of effective vaccines against both viruses is crucial for disease control. Recombinant viruses expressing heterologous antigens have shown promise as multivalent vaccine candidates. Unlike conventional PPRV vaccines, our recombinant RVFV-vectored vaccines offer a novel dual-protection strategy against RVF and PPR, combining safety, immunogenicity, and a DIVA strategy. Methods: Recombinant RVFVs (ZH548 strain) were generated to express either the hemagglutinin (H) or fusion (F) proteins from the PPRV strain Nigeria 75/1. The stability of these recombinant viruses was assessed through consecutive passages in cell culture. Immunogenicity studies were carried out in both mice and sheep to assess the induction of cellular and humoral immune responses capable of providing protection against RVFV and PPRV. These studies included intracellular cytokine staining (ICS), IFN-γ ELISAs, standard ELISAs for antibody detection, and virus neutralization assays. Results: The recombinant RVFVs expressing PPRV H or F proteins demonstrated stability in cell culture, maintaining high viral titers and consistent transgene expression over four passages. Immunization of mice resulted in the production of serum antibodies capable of neutralizing both RVFV and PPRV in vitro as well as cell-mediated immune responses specific to PPRV and RVFV antigens. In mice vaccinated with a high dose (10^5^ pfu), RVFV neutralizing titers reached ≥1:160 and PPRV neutralizing titers ranged from 1:40 to 1:80 by day 30 post-immunization. In sheep, neutralizing antibody titers against RVFV exceeded 1:160 as early as 2 days post-inoculation, while PPRV-specific neutralization titers reached up to 1:80 by day 21 in responsive individuals. In mice, administration of rZH548ΔNSs:F_PPRV_ elicited a detectable CD8+ IFNγ+ T-cell response against PPRV, with levels ranging from 1.29% to 1.56% for the low and high doses, respectively. In sheep, rZH548ΔNSs:F_PPRV_ also induced a robust IFNγ production against PPRV at 14 and 21 days post-infection (dpi). Conclusions: The successful generation and characterization of recombinant RVFVs expressing PPRV antigens demonstrate the potential of using rationally attenuated RVFV as a vector for multivalent vaccine development. Notably, the strategy proved more effective for the recombinant virus expressing the F protein, as it consistently induced more robust cellular and humoral immune responses. These results suggest that this approach could be a viable strategy for simultaneous immunization against Rift Valley fever and other prevalent ruminant diseases, such as peste des petits ruminants. Even though challenge studies were not performed in target species, the strong immune response observed supports including them in future studies.

## 1. Introduction

Rift Valley fever virus (RVFV) is an arbovirus transmitted by mosquitoes that causes severe disease outbreaks affecting humans and livestock in sub-Saharan Africa, the Middle East and the Indian Ocean islands [1]. Seasons of heavy rainfall favor the overgrowth of mosquito species capable of trans-ovarial transmission of the virus, which is associated with outbreaks in endemic countries. RVFV or *phlebovirus riftense* (fam. *Phenuiviridae*, class *Bunyaviricetes*) is an enveloped, single-stranded RNA virus characterized by a tri-segmented genome exhibiting both negative and ambisense polarity [2,3]. Although effective RVF vaccines are available across Africa, the unpredictable nature of outbreaks—often following prolonged inter-epizootic periods—continues to hinder the implementation of consistent annual vaccination programs [4,5]. One promising approach to mitigate this challenge is the use of multivalent vaccines, which can confer protection against a mandatory target disease while simultaneously offering immunity against other, more sporadically occurring, infections.

Peste des petits ruminants virus (PPRV) is a Morbillivirus of the *Paramyxoviridae* family that mainly affects domestic and wild ruminants. PPRV is the etiological agent of peste des petits ruminants (PPR or goat plague), a disease that causes substantial economic losses, with greater consequences in developing countries across Africa, the Middle East, and Asia where it is considered to be endemic. Mass vaccination is considered an effective way to control the spread of the disease, as was proven during the 2009–2011 campaigns in Morocco [6], which suppressed the virus until June 2015. Currently, live-attenuated PPR vaccines are being used in endemic regions [7] and inactivated vaccines are used in the rest of the countries [8]. Live-attenuated PPR vaccines induce a high level of protection for at least three years [9], but they carry the risk of reversion to virulence. On the other hand, inactivated vaccines are a safer choice, but they do not induce an immunity as good as live attenuated vaccines. Therefore, the development of improved vaccines that are safe and have DIVA properties (differentiation of infected from vaccinated animals) is a necessary step to improve the quality of the prophylactic measures. The development of DIVA vaccines would also facilitate the global PPR eradication program launched by the World Organization for Animal Health (WOAH, formerly OIE) in 2014 [10], since the DIVA strategy allows veterinary authorities to distinguish between naturally infected animals and those that are properly vaccinated. This capability supports targeted surveillance, safe trade practices, and outbreak containment, without penalizing vaccinated livestock. In this regard, recombinant vectored vaccines represent an ideal platform for DIVA implementation. In the case of PPRV, there are already several recombinant viral vectors, either poxvirus, herpesvirus, or adenovirus, that express PPRV proteins and can induce immunity and protection [11,12]. In addition to these well-established vaccine platforms, RVFV can also be engineered to support the expression of foreign antigens [13]. Previously, we also demonstrated that recombinant RVFV can express bluetongue virus antigens [14] that were able to elicit specific and protective immune responses in vivo. Similar approaches for other veterinary viral pathogens have been reported, such as a turkey herpesvirus vaccine against avian influenza and Newcastle disease [15] or a multivalent capripoxvirus-vectored vaccine [16].

In this work, we have reverse-engineered two recombinant RVFVs expressing either the PPRV hemagglutinin (H) or the fusion (F) proteins. The hemagglutinin (H) and fusion (F) proteins were selected as target antigens because they represent the major surface glycoproteins of PPRV and have been consistently implicated in protective immunity. The H protein is the primary target of virus-neutralizing antibodies and is essential for receptor binding, while the F protein mediates membrane fusion and entry into host cells. Specifically, our recombinant RVFV constructs lack the NSs gene and express epitope-tagged H or F PPRV antigens, allowing immunized animals to be serologically profiled using assays that differentiate vaccinated responses from those triggered by wild-type PPRV infection. Such DIVA features align with the requirements of the global PPR eradication campaign initiated by WOAH and FAO, and would facilitate disease tracking, post-vaccination monitoring, and international trade compliance. The expression of both proteins was able to induce an immune response in mice and sheep to levels that correlate with protection [17,18]. We therefore advocate the use of RVFV-vectored vaccines as a novel strategy to provide simultaneous immunity against PPR and RVF. The main benefit of using RVFV as a viral vaccine vector is that it could provide protection against RVF and other ruminant diseases, including peste des petits ruminants, thus becoming a very appealing strategy, particularly in the context of the current global eradication plans for PPR. This could be seen as an effective test to provide immunity against an endemic illness such as PPR while keeping animals protected against the unpredictable outbreaks of RVF.

## 2. Materials and Methods

### 2.1. Cells

The cell lines used for this study were HEK293T (human embryonic kidney 293 cells, ATCC CRL-3216), Vero (African green monkey kidney cells, ATCC CCL-81), and BHK-21 (baby hamster kidney fibroblasts, ATCC CCL-10). HEK293T cells were purchased from ATCC (Manassas, VA, USA), while Vero and BHK-21 cells were obtained from the cell repository maintained at CISA. Vero dog SLAM cells (VDS) cells were kindly provided by Dr S. Parida (IAH, Pirbright, UK). All cell lines were grown in Dulbecco’s modified Eagle medium (Invitrogen, Carlsbad, CA, USA) supplemented with 10% fetal bovine serum (FBS), 2 mM L-glutamine, 1% 100× non-essential amino acids, 100 U/mL penicillin, and 100 μg/mL streptomycin. All cells were incubated at 37 °C in the presence of 5% CO_2_.

### 2.2. Mice and Sheep

For the in vivo studies, 129/SvEv mice and autochthonous sheep of the Spanish *Churra* breed were used. Female mice, 8 weeks old, were obtained from our breeding facilities at the Department of Animal Reproduction (INIA-CSIC). Adult, 4-month-old female Churra sheep, of around 40 kg, were purchased from a local breeder. The sheep were housed and acclimatized for a week before the commencement of the experiment. During the whole experiment, the animals had free range of movements (no restrainers) with underlying stray. Water and food were supplied ad libitum. All experimental procedures were performed in accordance with EU guidelines (Directive 2010/63/EU) and protocols approved by the Animal Care and Biosafety Ethics Committees of Centro Nacional Instituto Nacional de Investigación y Tecnología Agraria y Alimentaria (INIA-CSIC) and Comunidad de Madrid (authorization decision PROEX 079.6/22).

### 2.3. Plasmid Construction and Recombinant RVFV Rescue

The two plasmids encoding the fusion (F) and hemagglutinin (H) gene sequences, with a size of 1758 bp and 1947 bp, respectively, from the Nigeria 75/1 PPRV strain (GenBank accession KY628761.1) were chemically synthesized by a commercial supplier (Biomatik Corporation, Cambridge, ON, Canada). Both sequences were flanked by specific restriction site sequences (AflII and NotI) and cloned into vector pCDNA3.1 to generate plasmids pCDNA3.1-F_PPRV_ and pCDNA3.1-H_PPRV_. Two restriction enzyme sites (XhoI and NcoI), internal to the 5′ and 3′ flanking sequences, were introduced for further subcloning. The 9-amino-acid short epitope V5 tag (IPNPLLGLD) sequence and the foot-and-mouth disease virus (FMDV) Cs8c1 site A sequence (recognized by the mAb SD6 [19]) were added at the 3′ end of each PPRV gene construct. The F and H genes (Figure 1) were extracted from the abovementioned plasmids through restriction enzymes and inserted in the previously digested pHH21-RVFV-vN_TCS plasmid [20]. The other rescue plasmids were already available and have been previously described [14,20].

For recombinant virus generation, a RNA pol I/II-based rescue system was used [20]. The pHH21 plasmids were used to provide the viral RNA segments (L, M, and S) of the ZH548 strain of Rift Valley fever virus (RVFV). Two additional pI.18 plasmids encoded S and L sequences, providing expression of the viral nucleoprotein N and the RNA-dependent RNA polymerase RdRp, respectively. In summary, a co-culture of BHK21 and HEK293T cells was transfected with pHH21-RVFV-vL, pHH21-RVFV-vM, and either pHH21-RVFV-vN-FPPRV or pHH21-RVFV-vN-HPPRV, along with pI.18-RVFV-L and pI.18-RVFV-N. Supernatants were collected at various time points following transfection and assessed for the induction of cytopathic effect (CPE) in Vero cell monolayers. When viral CPE was confirmed, viral stocks were generated by subculturing in Vero cells and used for the following experiments. The viruses generated lack the NSs gene (ΔNSs).

### 2.4. Viral Plaque Assay

Vero cells seeded in 12-well plates were infected with ten-fold serial dilutions of supernatants derived from previously infected cultures, using DMEM supplemented with 10% FBS. Following a 1 h incubation at 37 °C, the inoculum was removed and cells were overlaid with a semi-solid medium consisting of Eagle’s Minimum Essential Medium (EMEM) and 1% carboxymethylcellulose. Cultures were then incubated for 3 days at 37 °C for rZH548ΔNSs::F_PPRV_ and 5 days for rZH548ΔNSs::H_PPRV_. Post-incubation, cells were fixed with 10% formaldehyde, stained with 2% crystal violet, and viral titers were determined based on plaque counts at each dilution. Alternatively, plaques were visualized by immunostaining with the anti-RVFV nucleoprotein monoclonal antibody 2B1, following previously established protocols [21].

### 2.5. Indirect Immunofluorescence Assay

To confirm the expression of the heterologous genes following infection, Vero cells were inoculated with each recombinant virus at a multiplicity of infection (MOI) of 0.5. At 24 h post-infection, cells were fixed using a 1:1 methanol–acetone solution and incubated for 20 min at −20 °C. After removing the fixative, cells were incubated for 1 h at room temperature with PBS containing 20% FBS. A single PBS wash was performed prior to adding the primary antibodies, which were diluted 1:200 in PBS-FBS 20% for the polyclonal rabbit serum [21] and 1:500 for mouse monoclonal antibody targeting the V5 epitope tag (clone MCA 1360, Bio-Rad, Hercules, CA, USA). RVFV antigens were detected using the polyclonal rabbit serum, while the expression of H or F proteins was assessed with the antibody targeting the V5 epitope tag. Primary antibody incubation was carried out for 1 h at 37 °C or overnight at 4 °C. Cells were subsequently washed three times with PBS-T (PBS containing 0.05% Tween-20), followed by a 30 min incubation at room temperature with fluorescently labeled secondary antibodies: goat anti-mouse Alexa Fluor 488 or goat anti-rabbit Alexa Fluor 594 (Thermo, Waltham, MA, USA). Finally, cells underwent three additional PBS-T washes. Labeling of endoplasmic reticulum was achieved by mAb AHP516 (anti-calreticulin, Bio-Rad, Hercules, CA, USA). For FMDV detection, BHK-21 cells were infected with FMDV C-S8c1 and fixed 8 hpi and incubated with either mice or sheep sera. Nuclei were visualized by DAPI staining (Thermo). Stained cells were visualized and images were obtained using a Zeiss LSM880 confocal laser microscope (Gmbn, Oberkochen, Germany).

### 2.6. Western Blot and Immunoblotting

Western blot analysis was carried out to evaluate the kinetics of vaccine antigen expression and to indirectly assess their genetic stability. For the kinetics of the antigen expression, monolayers of Vero cells were infected with rZH548ΔNSs::H_PPRV_ or rZH548ΔNSs::F_PPRV_. After 24, 48, and 72 h, cells were washed twice with PBS and lysed in SDS-PAGE sample buffer (Bio-Rad, Cat#1610747, Hercules, CA, USA) with reducing agent 2-mercaptoethanol. Proteins separated by SDS-PAGE were transferred onto nitrocellulose membranes (Amersham, Cytiva, Chicago, IL, USA) using wet electroblotting at 100 V for 1 h. The transfer was performed using a buffer containing 25 mM Tris, 192 mM glycine, and 20% methanol, prepared in ultrapure water. All steps were carried out at room temperature under constant voltage conditions. The membranes were incubated for one hour with 5% skim fat milk in TBS-T (TBS supplemented with 0.05% Tween-20). Upon blocking, an anti-V5 epitope specific mAb (MCA1360, Bio-Rad, Hercules, CA, USA) was used diluted 1:1000 in TBS-T.

Membranes were subsequently washed three times with TBS-T, followed by a 1 h incubation at room temperature with an anti-mouse polyvalent immunoglobulin conjugated to HRPO (Sigma-Aldrich, St. Louis, MO, USA) diluted 1:1000 in TBS-T. After additional three washes with TBS-T, membranes were treated with an enhanced chemiluminescence substrate (ECL Plus, GE Healthcare 28980926, Chicago, IL, USA), and signals were visualized using a ChemiDoc Imaging System (Bio-Rad, Hercules, CA, USA). All washing steps were performed at room temperature, with each wash lasting for 30 min at a constant agitation speed of 40 rpm.

For the genetic stability assessment, supernatants from Vero cells infected with each recombinant virus were collected after the first through fifth serial passages. Viral titers were determined, and fresh Vero cell monolayers were infected at a multiplicity of infection (MOI) of 1 to evaluate heterologous expression consistency across passages. After 24 h, cells were collected and analyzed by SDS-PAGE as explained above. Upon blocking, the membranes were incubated with the anti-V5 epitope mAb (for transgene expression detection) and the 2B1 anti-RVFV-N mAb [21] both diluted 1:1000 in 5% milk-TBS-T (for infection reference normalization). Membranes were subsequently washed three times with TBS-T, followed by a 1 h incubation at room temperature with an anti-mouse polyvalent immunoglobulin conjugated to HRPO (Sigma-Aldrich, St. Louis, MO, USA) diluted 1:1000 in TBS-T. Subsequent steps—including washing, secondary antibody incubation, chemiluminescent detection, and imaging—were performed as described for the antigen expression assay.

### 2.7. Mouse and Sheep Immunization and Sampling

Five groups of 129/SvEv mice (n = 5) were inoculated intraperitoneally with 10^5^ (high dose, HD) or 5 × 10^2^ (low dose, LD) pfu/mouse of either rZH548ΔNSs::H_PPRV_ or rZH548ΔNSs::F_PPRV_. One group of mice was kept as mock-inoculated control. Mice were monitored daily after inoculation and bled at different time points (2, 3, 4, 5, 14 and 21 days post-infection) to check for viremia and specific antibodies induction. Four weeks after inoculation (immunization), the mice were sacrificed and their spleens were collected for intracellular cytokine staining assay (ICS).

Three groups of four sheep were immunized with 10^7^ pfu/sheep of rZH548ΔNSs::F_PPRV_ or 10^5^ pfu/sheep of rZH548ΔNSs::H_PPRV_ or were kept unimmunized. After immunization, animals were monitored and bled at different time points (7, 14 and 21 dpi) to evaluate the humoral and cellular immune response induced by the recombinant virus.

### 2.8. Anti-PPRV IgG ELISA

IgG anti-PPRV detection was performed as described previously [22]. Shortly, ELISA plates (Maxisorp, Nunc, Rochester, NY, USA) were coated and incubated overnight at 4 °C with 10^4^ pfu per well of saccharose-cushion-purified PPRV Nigeria 75/1. Washes and blockage were performed with 0.1% Tween in PBS and 10% FBS in PBS, respectively. PPRV-specific IgGs were detected with a secondary donkey anti-sheep IgG horseradish-peroxidase-conjugated antibody (Serotec, Kidlington, UK) diluted 1:6666 in PBS + 0.5% FBS or HRPO-conjugated goat anti-mouse IgG antibody. Detection was performed using the Liquid Substrate System (Sigma-Aldrich, St. Louis, MO, USA), based on 3,3′,5,5′-Tetramethylbenzidine (TMB) as the developing agent. Reactions were stopped by adding 3 M sulfuric acid, and optical density (OD) was measured at 450 nm using an ELISA FLUOstar Omega plate reader with the MARS Data Analysis Software v2.10 (BMG Labtech, Ortenberg, Germany). All measurements were performed in triplicate and deemed valid when the standard deviation remained below 10% of the mean. Mouse sera were tested in ELISA at 1:100 dilution. Anti-PPRV IgG titers in sheep serum were defined as the dilution at which the OD value doubled that of the corresponding pre-immune serum from the same animal. Titers were calculated using linear regression analysis of OD values plotted against serum dilutions.

### 2.9. Anti-RVFV-N IgG and Competition ELISAs

ELISA plates (Maxisorp, Nunc, Roskilde, Denmark) were coated and incubated overnight at 4 °C with 50–100 ng/well of purified recombinant RVFV-N protein diluted in carbonate/bicarbonate buffer at pH 9.6. Washes and blockage steps were performed with PBS 0.1% Tween (PBS-T) and 5% skim milk in PBS, respectively. Sera from inoculated mice were diluted 1:100 in PBS supplemented with 5% skim milk prior to addition to the plate. After three consecutive washing steps, bound antibodies were detected using an HRPO-conjugated goat anti-mouse antibody (Bio-Rad, Hercules, CA, USA) diluted 1:1000. ELISA signaling and measurements were performed as described above. The competition ELISA for the detection of RVFV nucleoprotein N-specific antibodies in serum was carried out by means of IDVet RVF screen Kit (IDVet, Grabels, France).

### 2.10. RVFV Neutralization Test

Sera from vaccinated mice or sheep were heat-inactivated at 56 °C for 30 min and subsequently subjected to two-fold serial dilutions in DMEM, starting at a 1:10 dilution. Each diluted sample was mixed in equal volume (50 μL) with RVFV-MP12 viral stock containing 10^3^ plaque-forming units (pfus) per well. Following a 1 h incubation at 37 °C, the mixtures were transferred onto Vero cell monolayers previously seeded in 96-well plates and incubated at 37 °C with 5% CO_2_. After a 3-day incubation period, cells were fixed and stained using 2% crystal violet in 10% formaldehyde. Neutralizing antibody titers were defined as the highest serum dilution capable of reducing cytopathic effect (CPE) by 50%.

### 2.11. PPRV Neutralization Test (VNT)

Sheep or mouse serum samples were tested for the presence of neutralizing antibodies as described previously [23,24]. Briefly, Nigeria 75/1 PPRV stock (100 pfu/well) was incubated with serial dilutions of inactivated (30 min 56 °C) sheep sera for 1 h at 37 °C in 96-well plates (M-96). Then, VDS cells (2 × 10^4^ cells/well) were added to the mixtures and incubated at 37 °C and 5% CO_2_. After 5 days, they were fixed with 2% formaldehyde and stained with crystal violet. All dilutions were performed in duplicate, and the neutralization titer is expressed as the reciprocal of the highest dilution of sera at which virus infection is blocked.

### 2.12. Intracellular Cytokine Staining

Collected spleen cells were stimulated with a peptide pool of three F- and two H-specific peptides from PPRV Nigeria 75/1 (10 μg/mL each) (GenBank #CAJ01699.1 and #CAJ01700.1, respectively) (Table 1), as previously described [25], with ConA mitogen (8 μg/mL) or were left untreated in 10% FBS-supplemented RPMI 1640 medium for 6 h. Following stimulation, spleen cells were washed and stained with surface markers prior to fixation and permeabilization for intracellular labeling using specific fluorochromes. Immunostaining was performed using the following fluorochrome-conjugated anti-mouse antibodies: anti-CD8-PE, anti-IFNγ-FITC, and anti-CD4-APC (Miltenyi, San Jose, CA, USA). Flow cytometric data were acquired using a Cube 8 cytometer (Sysmex Spain S.L., Sant Just Desvern, Spain) and analyzed with FlowJo software version X0.7 (Tree Star, Ashland, OR, USA). A total of 5 × 10^4^ events were collected within the lymphocyte gate.

### 2.13. Detection of IFNγ in Sheep Plasma Samples by Capture ELISA

IFNγ capture ELISA was carried out as previously described [26]. Briefly, 100 µL of heparinized blood, collected at different time points before and after inoculation from each animal, were incubated in multi-well plates with a peptide pool of three F-specific and two H-specific peptides from PPRV Nigeria 75/1 (10 μg/mL each) (Table 1) or purified recombinant RVFV Gn or N proteins for 48 h. Following centrifugation, plasma samples were harvested and stored at −80 °C until analysis. For each time point, 50 μL of plasma was subjected to capture ELISA. After performing the washing steps, biotinylated anti-bovine IFNγ (clone MT307, Mabtech, Nacka Strand, Sweden) was added to the wells. Immunocomplexes were detected using streptavidin-conjugated HRPO (Becton-Dickinson, Franklin Lakes, NJ, USA), followed by development with TMB peroxidase substrate (Sigma-Aldrich, St. Louis, MO, USA). Absorbance was measured at 450 nm using an automated microplate reader (BMG Labtech, Ortenberg, Germany).

### 2.14. Statistical Analyses

Statistical analyses were conducted using GraphPad Prism version 10 (GraphPad Software, San Diego, CA, USA). Group comparisons for antibody levels, T-cell responses, serum biochemical markers, fever, and viremia were evaluated using analysis of variance (ANOVA). Statistical significance was defined as *p* < 0.05 for all comparisons.

## 3. Results

### 3.1. Recombinant RVFV Generation and Heterologous Protein Expression

Recombinant RVFVs (rRVFV) expressing F or H proteins were rescued using the RNA pol I/II-based rescue system as briefly described in Section 2. Two simultaneous attempts were made to rescue the recombinant viruses. Both attempts were successful, so that recombinant viruses expressing either the F or the H proteins could be serially propagated. As the expression of the heterologous proteins was high and homogeneous, no further cloning was necessary. The growth kinetics of the rescued viruses was analyzed in cell culture, and differences in yield were observed; the final titers reached by rZH548-ΔNSs::H_PPRV_ were lower by nearly two logs (Figure 2A). Accordingly, rZH548-ΔNSs::H_PPRV_ and rZH548-ΔNSs::F_PPRV_ viruses differed in plaque phenotypes (Figure 2C,D), with those of rZH548-ΔNSs::H_PPRV_ being smaller and almost unappreciable, consistent with a lower cytopathogenicity.

The kinetics of expression of PPRV F and H transgenes upon infection in Vero cells was also confirmed by Western blot analysis (Figure 2B) of cell lysates collected at 24, 48, and 72 h post-infection. Using the anti-V5tag monoclonal antibody, both proteins were clearly detected at all time points analyzed. Three protein bands were detected in rZH548-ΔNSs::H_PPRV_-infected cells. A major protein species corresponded well with the expected size of the full-length H protein, and the two proteins with a higher molecular weight could result from different degrees of glycosylation (although this point was not further addressed). On the other hand, in rZH548-ΔNSs::F_PPRV_-infected cells, two main discrete bands were clearly detected, which could correspond with the unprocessed (F0) and processed (F1) forms of the F protein, consistent with a proteolytic event carried out by cellular proteases as described in other paramyxovirus species.

After rescue, the heterologous H and F gene expression was also analyzed by immunofluorescence in cell cultures. rRVFV-infected Vero cells were detected with a polyclonal anti-RVFV antibody, while expression of either PPRV H or F was detected using the anti-V5tag monoclonal antibody specific for the epitope inserted at the C-terminus of each recombinant gene. Figure 3A shows a localized staining pattern for both PPRV proteins accumulating close to the cell nucleus. Interestingly, the F-specific signal was also detected lining the cellular membranes. Using an anti-calreticulin antibody we could confirm the subcellular localization of both proteins in association with ER, indicating their interaction with the cell secretory pathway (Figure 3B). As expected, both H and F genes were also detected by immunofluorescence using the SD6 mAb specific for the FMDV tag included in the constructs.

### 3.2. Analysis of the Stability upon Cell Passage of rZH548-ΔNSs::H_PPRV_ or rZH548-ΔNSs::F_PPRV_

Since both foreign PPRV genes introduced in the genome of the recombinant viruses altered the normal size of the small (S) RNA genomic segment, we tested the stability of both viruses by analyzing the expression level of both transgenes after serial cell culture passages (Figure 4A). Cell extracts from infected cell cultures, collected at different passage number, were analyzed by Western blot. After four consecutive passages, both proteins could still be detected. The overall H expression levels were clearly lower than F expression levels, being almost undetectable at passage four. We used RVFV nucleoprotein N expression levels to normalize the level of expression of each PPRV transgene. Densitometric H:N and F:N ratios were similar (Figure 4B), decreasing upon cell passages. The H:N ratio decreased faster, with a value of 0.45 by passage 4 and H expression was not detected by passage 5, while the F:N ratio at passage 4 was still 0.70 and, by passage 5, expression levels were very low but still detectable. These data confirm that the recombinant virus expressing the heterologous antigens could be maintained at least during four consecutive passages, allowing the production of larger stocks suitable for in vivo immunizations. We did not investigate the fate of the virus beyond passage 5.

### 3.3. Analysis of Immune Responses in Vivo

Since RVF viruses lacking NSs are fully attenuated in immunocompetent hosts, they can be used to induce a safe immune response. To test the immunogenicity elicited by the recombinant viruses encoding H or F PPRV proteins, 129/SvEv mice were inoculated intraperitoneally with a low (500 pfu) or high dose (10^5^ pfu) of each virus (rZH548-ΔNSs::H_PPRV_ and rZH548-ΔNSs::F_PPRV_). At early times post-inoculation (2–3 dpi), the mice were bled and RNA was extracted from blood samples to analyze the presence of viremia. As expected, no viral RNA was detected in any of the analyzed blood samples. These data, together with the absence of clinical signs and weight loss upon inoculation, confirm that neither H nor F expression alters the virus phenotype, with both viruses remaining fully attenuated in immunocompetent mice.

At 30 days post-immunization, animals were bled and serum samples were taken to analyze the levels of antibodies against both RVFV and PPRV. Higher levels of anti-N antibodies were detected in the high-dose immunization groups (Figure 5A). In the low-dose immunization groups, lower levels of anti-N antibodies were detected, with even two negative animals and two more at the sensitivity limit in the rZH548-ΔNSs::H_PPRV_ group. A semi-purified PPRV antigen–lysate-based ELISA showed that all immunized animals except one had detectable antibodies, although all animals in the low-immunization groups presented levels very close to the sensitivity limits (Figure 5C), confirming a dose-dependent detection of antibodies. These data on successful immunization were also confirmed by detection of antibodies against foot-and-mouth disease virus (FMDV) in infected cell cultures, since both transgenes encode the B-cell antigenic site A of FMDV C-terminally tagged (Figure 5E). Similarly, all immunized animals, except two mice from the group inoculated with the lower dose of rZH548-ΔNSs::H_PPRV_, showed detectable RVFV neutralizing antibodies in vitro, with higher levels observed in those mice vaccinated with 10^5^ pfu, consistent with dose-dependent induction (Figure 5B). When it comes to neutralizing antibodies against PPRV, a similar dose-dependent pattern is observed (Figure 5D). However, these serum antibodies failed to neutralize an FMDV infection in vitro.

### 3.4. rZH548-ΔNSs::F_PPRV_ Induces a Cellular Response in Immunized Mice

Since vaccination of animals with viral vectors expressing H or F PPRV proteins induced cellular immune responses [25], we wanted to determine if the vaccination with rZH548-ΔNSs::H_PPRV_ or rZH548-ΔNSs::F_PPRV_ would also trigger T-cell responses specific for H and F proteins. To monitor this response, spleen cells from immunized mice were stimulated with a pool of H and F peptides (Table 1). As can be observed in Figure 6, a remarkable level of CD8+ T-cell IFN-γ production was detected upon stimulation with the F peptide pool in spleen cells of mice immunized with rZH548-ΔNSs::F_PPRV_, with a higher response induced in the high-dose immunization group. rZH548-ΔNSs::H_PPRV_ did not trigger such a level of specific CD8-T-cells, although they appeared over background (control) levels. For both viruses, the levels of stimulated CD4-T-cells showed no significant differences.

### 3.5. Assessment of Infectivity and Immunogenicity of Recombinant rZH548-ΔNSs::H_PPRV_ and rZH548-ΔNSs::F_PPRV_ in Sheep

The previous results obtained in mice led us to test the attenuation and immunogenicity of the rRVFVs-PPRV in sheep. Therefore, we planned a pilot vaccination experiment using two groups of n = 4 sheep each, at the maximum titer reached by each recombinant virus (i.e., one group was vaccinated with 10^5^ pfu of rZH548ΔNSs::H_PPRV_ and the other with 10^7^ pfu of rZH548-ΔNSs::F_PPRV)_. After inoculation, the animals were monitored, and no clinical signs appeared at any time. Animals were also bled to check for the presence of viraemia, and again, virus was not detected at any time post-immunization, thus confirming the safety of these vaccines.

The kinetics of the RVFV-specific neutralizing antibody response revealed detectable serum levels as early as 2 days post-inoculation (dpi), with titers peaking at 14 dpi and remaining relatively stable through day 21 post-vaccination (Figure 7A). Although the levels of anti-RVFV neutralizing antibodies induced by the two viruses were equivalent, surprisingly, the level of anti-RVFV-N antibodies detected by ELISA (Figure 7B) was slightly higher, although not significantly, in sheep vaccinated with rZH548ΔNSs::H_PPRV_. Notably, this improvement was observed in the group that received the lower inoculation dose, further supporting the idea that a reduced dose may also be optimal for inducing a robust immune response. Neutralizing antibodies against PPRV appeared by day 14 post-vaccination in sheep immunized with rZH548ΔNSs::F_PPRV_ and by day 21 post-vaccination in sheep immunized with rZH548ΔNSs::H_PPRV_ (Figure 7C). One out of four sheep vaccinated with rZH548ΔNSs::F_PPRV_ and two out of four sheep vaccinated with rZH548ΔNSs::H_PPRV_ did not show detectable neutralizing antibodies against PPRV at any time post-vaccination. However, all animals vaccinated with rZH548ΔNSs::H_PPRV_ or rZH548ΔNSs::F_PPRV_ showed PPRV-specific antibodies from day 14 post-vaccination, as determined by ELISA (Figure 7D). As for those of mice, sheep sera were also capable of specifically detecting FMDV-infected cells (Figure 7E). However, these were not able to neutralize the virus in vitro.

Cellular immune responses in both vaccine groups were assessed at multiple time points following challenge, using an IFNγ capture ELISA (Figure 8). Increased levels of IFNγ were detected in plasma samples from sheep immunized with rZH548-ΔNSs::F_PPRV_ when samples from days 14 and 21 post-immunization were re-stimulated with a pool of F peptides, or with purified RVFV N and Gn. In contrast, samples from sheep immunized with a low dose of rZH548-ΔNSs::H_PPRV_ only showed IFNγ induction when were re-stimulated with recombinant RVFV N or Gn.

## 4. Discussion

Vaccination is a potential means of controlling RVF in ruminants. From a One Health perspective, immunizing ruminants would reduce the impact of this disease on humans. However, despite the availability of effective live-attenuated RVFV vaccines, implementing a vaccination policy in most African countries is challenging due to the prolonged inter-epizootic periods between RVF outbreaks. Vaccination is usually only adopted after epizootic outbreaks have occurred, often too late to prevent the virus being transmitted between animals and humans. This makes it difficult for herd owners to perceive RVF vaccination as a necessity. One potential solution is to integrate RVF prevention activities with other vaccination programs for small ruminants. Implementing a routine preventive vaccination program would be highly beneficial, as it would maintain protective levels of immunity and prevent the rapid spread of an outbreak. Co-immunization of livestock with vaccines against RVF and other prevalent ruminant diseases requiring regular vaccination campaigns, such as PPR, could be an effective strategy, particularly in the context of the current global PPR eradication campaign.

The use of viral vector vaccines expressing PPRV H and F antigens induces a humoral and cellular response that is able to protect animals against a PPRV challenge [25,27,28]. Similarly, the expression of RVFV glycoproteins Gn and Gc is sufficient to ensure protective immune responses in ruminant species [27,29]. Different viral vectored vaccines have been proposed for both viruses such as adenovirus platforms [30,31], poxvirus [32,33,34], and herpesvirus [35,36] among them. Combining PPR and RVF vaccines has been also proposed as an easy means to improve vaccination coverage and efficacy [37]. Here, we propose the use of attenuated RVFV as a viral vector encoding heterologous vaccine antigens, as shown in previous studies [13,14]. The attenuation of the virus is achieved by deleting the virulence-associated NSs gene, which is non-essential for the productive replication of the virus in cell culture [38]. NSs gene deletion generates an insertion site that, in agreement with the previously demonstrated genome plasticity of RVFV [39], theoretically allows for the insertion of any gene of interest. Inoculation of adult animals with these recombinant, NSs deleted, viral vaccines expressing tagged PPRV H and F genes was safe and induced no adverse effects. Animals monitored after inoculation showed no viremia, fever, or any other biochemical or hematological alterations.

The expression of both H and F proteins by means of the RVFV vector was highly satisfactory. Both full-length proteins were detectable in association with intracellular membranes, suggesting that proper intracellular trafficking through the cell secretory pathway and post-translational protein modifications took place. Both heterologous antigens were stable in vitro, with little or no signs of degradation during the time course of infection and upon serial propagation. Moreover, the PPRV F protein was proteolytically processed as expected, in agreement with the protein fragment sizes observed in the Western blot analysis. The molecular weights of these processed forms fit well with what has been described for measles virus (MeV) envelope glycoproteins and other morbilliviruses (around 60kDa and 41kDa for F0 and F1, respectively) [25,40,41]. Therefore, it is reasonable to assume that the expression of both proteins mimics that of an authentic PPRV infection, thus providing similar antigen-presenting properties and facilitating the induction of adequate immune responses. The stability of the heterologous antigens remains a key concern in the development of this vaccine platform. Variability in expression levels across serial passages, particularly for the H protein, highlights the need for further optimization. We are currently conducting follow-up studies to investigate the underlying causes of instability and explore strategies to improve transgene integrity and sustained expression. In our study, we have shown that delivery of either H or F antigens by using a recombinant RVFV is able to elicit presumably protective B- and T-cell responses including antigen presentation via the MHC class I pathway, thereby promoting cytotoxic CD8^+^ T-cell activation [42]. In contrast, the hemagglutinin (H) protein mediates receptor attachment and is predominantly exposed on the viral surface, making it a key target for neutralizing antibodies and favoring a humoral immune profile [43]. It is known that both responses are useful in protection, since the induction of IgG and neutralizing antibodies is necessary to avoid or prevent reinfections, and a good cellular response is key to rapidly target and eliminate infected cells. After a single immunization with rZH548-ΔNSs::H_PPRV_ or rZH548-ΔNSs::F_PPRV_, both non-neutralizing and neutralizing PPRV-specific antibodies were detected in mice and sheep sera, the latter with titers that could be considered protective according to the reported literature data [22] and therefore predictive of vaccine efficacy [24]. In both animal models, anti-RVFV neutralizing titers were also produced at protective levels, as previously reported [44]. As was the case with the previously mentioned virus, antibodies against FMDV were also elicited, though these were non-neutralizing in both animal models. The lack of FMDV-neutralizing capabilities of these sera may suggest that the sequence of the site A fused to either the recombinant H or F proteins is not properly exposed, as it is in the native conformation of FMDV capsids. If this is the case, redesigning the inserted site A sequence to make it similar to that of FMDV particles would be an interesting improvement to add additional protective features to these recombinant viruses.

Our results warrant further experiments to test the in vivo protective capacity against PPRV challenge in either sheep or goats. An efficacy/challenge study in target hosts was not attempted in this study because the immune responses against PPRV in sheep were not homogeneous (three out of eight vaccinated animals did not show detectable serum neutralizing antibodies). Three out of eight vaccinated sheep did not develop detectable serum neutralizing antibodies, with two of these belonging to the rZH548-ΔNSs::HPPRV group. Therefore, this lack of response could be explained by the low inoculation dose administered to the sheep, which was as low as the low dose used in the mouse experiment. On the other hand, since the inoculation of 10^7^ pfu/sheep with rZH548-ΔNSs::F_PPRV_ was completely safe in healthy animals, with no evidence of viremia, it may be possible to test a higher dose, e.g., 10^8^ PFU per animal, to guarantee more uniform immunity in all vaccinated individuals, especially considering that one animal in this group also failed to respond. It is important to note that the aim of this experiment was not to compare the level of protection induced by the different antigens, but rather to determine whether both were capable of eliciting an immune response in the animals. Remarkably, our data clearly show that a higher inoculation dose elicits stronger and more consistent protective immune responses. This finding strongly supports the use of increased doses in future experiments to achieve robust and homogeneous immunity. Such an approach would also allow for a more accurate evaluation of whether one antigen is more suitable for vaccine development.

Beyond the dose effect, another possible explanation for the lower and delayed anti-PPRV-H antibody responses could be related to the lower protein expression level, smaller plaque phenotype, and slower growth kinetics exhibited by rZH548-ΔNSs::H_PPRV_ in Vero cells (Figure 1 and Figure 2). Since infection with rZH548-ΔNSs::H_PPRV_ was less cytopathogenic than that of rZH548-ΔNSs::F_PPRV_, it could be suggestive of an interfering effect of the PPRV hemagglutinin with the normal RVFV cell cycle. Attempts to grow rZH548-ΔNSs::H_PPRV_ to reach higher titers in other, more permissive cell lines are in progress.

Currently, new-generation PPR vaccines are sought to be also capable of inducing a cellular immune response because it is known that T-cell activation contributes to protection against infection and confers a cross-protective immunity [45]. Immunization with both recombinant viruses also elicited PPRV-specific CD4+ and CD8+ T-cell responses, reflecting the potential of protecting against multiple PPRV strains. Although differences were not significant, lower induction of cellular and humoral responses were detected after inoculation with rZH548-ΔNSs::H_PPRV_. These results are consistent with the reduced humoral immune response likely caused by the lower inoculation dose and possibly by the lower expression levels of the hemagglutinin antigen, as previously discussed.

Although it is not yet clear whether combining H and F expression will improve protection, some studies suggest that using both antigens can induce better protection than using one antigen alone [18,46]. Therefore, the combination of both RVFV-vectored PPR vaccines is another aspect to consider for future efficacy experiments. Having proven these strategies successful, combining recombinant RVFV vaccines could provide a new research opportunity to test vaccine cocktails for simultaneous immunization against multiple ruminant diseases. Previous studies have shown that a rRVFV expressing either bluetongue virus VP2 or NS1 antigens can elicit protective immune responses against a BTV-4 challenge [14]. Therefore, it would be feasible in the future to provide immunity against RVFV and other small ruminant diseases by combining rRVFV vaccines in a single-dose approach.

While our findings show that recombinant RVFV vectors expressing PPRV antigens are safe and immunogenic in both mice and sheep, several other aspects remain to be addressed in future studies. First, this work evaluates immune responses only during the acute post-vaccination phase (up to 30 days) and, therefore, does not provide data on long-term protection or immunological memory. Monitoring antibody and T-cell durability beyond this time frame will be essential to determine the need for booster doses and the true value of a single-shot strategy. Second, although immunogenicity results are promising, we did not assess protective efficacy in a live virus challenge model. As previously discussed, a critical next step will be to repeat the immunization in sheep using increased and equivalent doses of both recombinant inocula. This will allow for a direct comparison of the immune responses elicited by each antigen under optimized conditions and help determine whether one construct offers superior protection or whether their combination may be more effective. Lastly, the DIVA properties of this vaccine have not yet been experimentally confirmed. However, the design of the recombinant RVFV vectors, which do not express the PPRV nucleoprotein (N) [47], supports DIVA functionality in principle, since current serological diagnostics for PPRV infection rely predominantly on anti-N antibody detection. By excluding this major diagnostic antigen from our vaccine constructs, vaccinated animals should test negative in standard N-based ELISAs, while naturally infected animals would remain positive. This theoretical advantage requires future validation in field conditions to confirm the accuracy and reliability of differentiation in vaccinated populations, particularly in endemic regions where both exposure and immunization may coexist.

## Figures and Tables

**Figure 1 vaccines-13-01039-f001:**
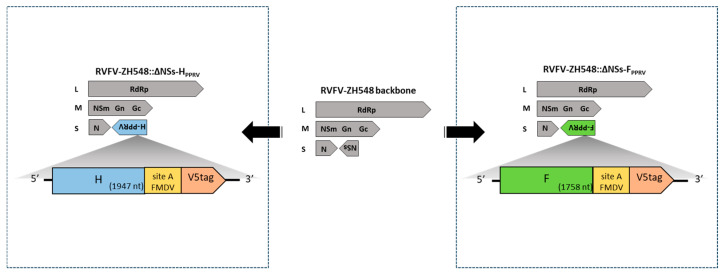
Schematic representation of the genomic structure of the rRVFV-PPRV rescued by reverse genetics system. The foreign sequences to be inserted by means of two pHH21-vN constructs are detailed (not to scale). The V5 tag epitope and the FMDV Cs8c1 site A sequence (recognized by the SD6 mAb) were added at the 3′ end of each PPRV ORF for detection purposes. Inverted writing denotes the coding orientation with respect to the nucleoprotein ORF in the viral genome. Nucleotide numbers indicate the size of the inserted PPRV sequences.

**Figure 2 vaccines-13-01039-f002:**
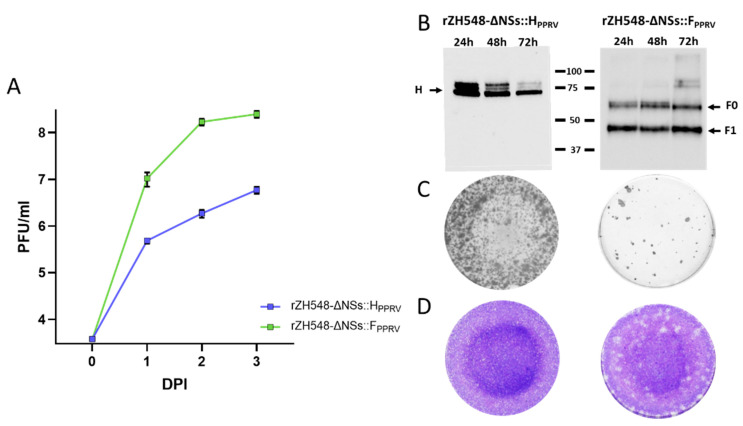
Characterization of rescued RVFVs. (**A**) Growth kinetics of rZH548-ΔNSs::H_PPRV_ and rZH548-ΔNSs::F_PPRV_. Vero cells were infected with an MOI of 0.01, and supernatants were harvested at the indicated time points (days post-infection). Viral titers (mean ± SD) were determined by plaque assay upon infection of Vero cells with serial dilutions of the supernatants. Titrations were performed in duplicates. (**B**) Western blot analysis of rZH548-ΔNSs::H_PPRV_- or rZH548-ΔNSs::F_PPRV_-infected Vero cells at different times post-infection. Anti-V5tag mAb was used for the detection of either H or F recombinant proteins. The arrows point to the detection of F0 (unprocessed), F1 (processed), and H proteins. (**C**,**D**) Plaque phenotypes of the rescued recombinant viruses. Cell lysis was visualized at 3–5 days post-infection: immunostaining with anti-N mAb 2B1 (**C**) or Crystal violet staining (**D**).

**Figure 3 vaccines-13-01039-f003:**
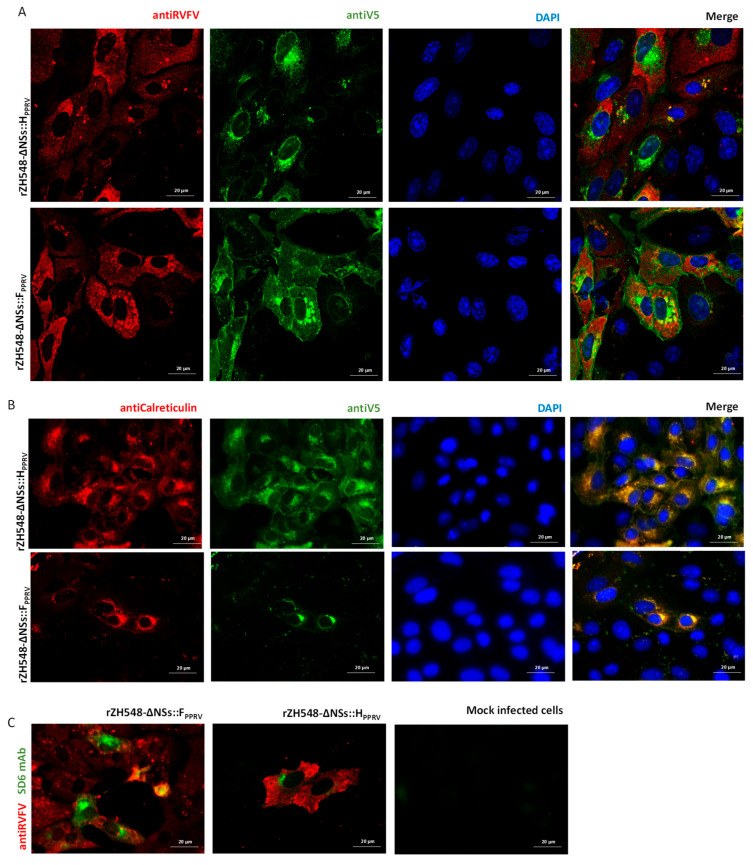
Detection of recombinant PPRV transgene expression in RVFV-infected Vero cell cultures. (**A**) Indirect immunofluorescence assay (IFA) of rescued recombinant RVFVs in infected Vero cells. RVFV antigens were visualized using a rabbit polyclonal anti-RVFV serum (red fluorescence), while expression of the PPRV H and F proteins was detected using a mouse anti-V5 epitope antibody (green fluorescence). (**B**) Representative images showing the subcellular localization of the recombinant H and F proteins in infected cells. H and F antigens were detected with anti-V5tag antibody (green fluorescence) and anti-calreticulin (red fluorescence) used to localize ER. DAPI staining was used to show cell nuclei. Magnification 63×. (**C**) Detection of the FMDV site A tag by SD6 mAb (green fluorescence) and RVFV antigens (red fluorescence).

**Figure 4 vaccines-13-01039-f004:**
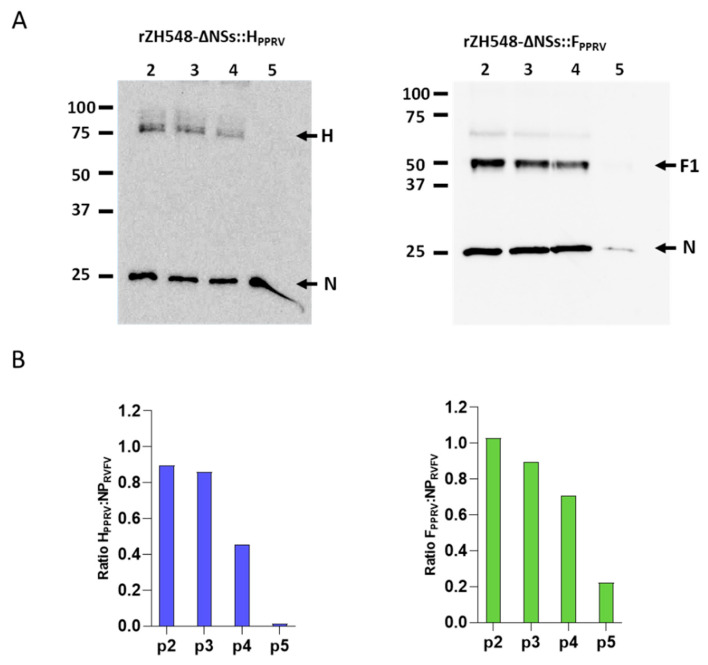
Stability of recombinant antigen expression. (**A**) Expression upon cell passage of rZH548-ΔNSs::H_PPRV_ or rZH548-ΔNSs::F_PPRV_. Total cell extracts were separated by SDS-PAGE and subjected to Western blot transfer. Anti-V5tag mAb was used for the detection of either H or F recombinant proteins. Anti-N mAb 2B1 was used to detect the expression of RVFV nucleoprotein N. Molecular mass in kDa. (**B**) Relative detection of H/N or F/N ratios derived from densitometric WB values at different passages.

**Figure 5 vaccines-13-01039-f005:**
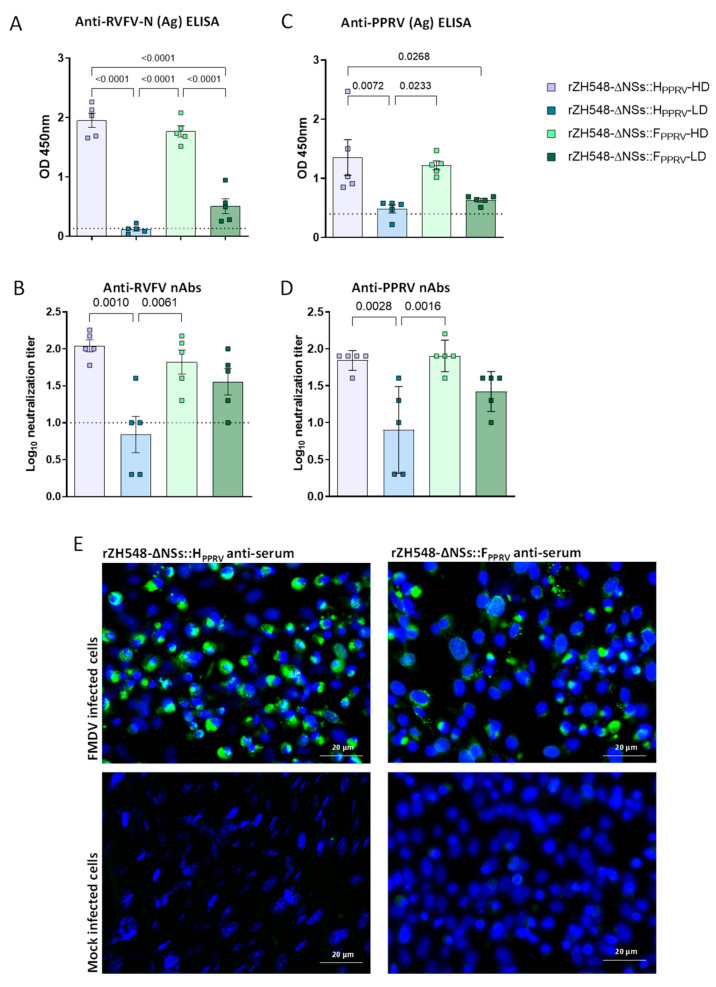
Humoral responses following a single-dose inoculation with rZH548-ΔNSs::H_PPRV_ or rZH548-ΔNSs::F_PPRV_ in mice. (**A**) ELISA-based detection of anti-nucleoprotein (N) antibodies in serum samples collected 30 days post-inoculation with two different doses of the recombinant viruses. Plates were coated with purified recombinant N protein, and serum was diluted 1:100. (**B**) Evaluation of RVFV-specific neutralizing antibody responses induced by immunization. Neutralization titers are expressed by the log_10_ of the highest serum dilutions reducing CPE by 50%. Dotted line indicates the sensitivity threshold (1/10 dilution). (**C**) Detection of PPRV-specific serum antibodies by ELISA. ELISA plates were coated with semi-purified PPRV (10^4^ pfu equivalents). OD values correspond to a serum dilution of 1/100. (**D**) Neutralizing antibody induction against PPRV after immunization. Neut. titers expressed as the highest dilution of serum blocking CPE (100%). Horizontal dotted lines represent assays’ sensitivity limits. (**E**) PPRV-H- or F-transfected and FMDV-infected BHK-21 cells were detected using sera from mice inoculated with 10^5^ pfus of rZH548-ΔNSs::H_PPRV_ or rZH548-ΔNSs::F_PPRV_. HD: high dose, 10^5^ pfu/mL; LD: low dose, 500 pfu/mL. Data (**A**–**D**) are presented as mean ± standard error of the mean (SEM). Statistical comparisons (**A**–**D**) were performed using one-way ANOVA.

**Figure 6 vaccines-13-01039-f006:**
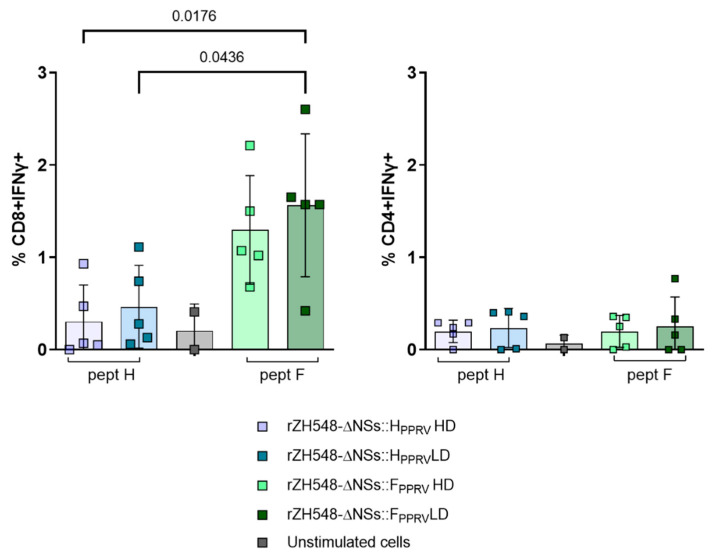
Detection of transgene-specific cellular responses. Intracellular cytokine cell staining (ICCS). Percentage of CD8+ and CD4+ IFNγ+ T-cells after re-stimulation with a pool of H or F peptides. Data are presented as mean percentages of IFNγ+ T-cells normalized to RPMI medium stimulated cells for each mouse. HD: high dose, 10^5^ pfu/mL; LD: low dose, 500 pfu/mL. Data are presented as mean ± standard error of the mean (SEM). Statistical comparisons were performed using one-way ANOVA. Individual data points are plotted to represent biological replicates.

**Figure 7 vaccines-13-01039-f007:**
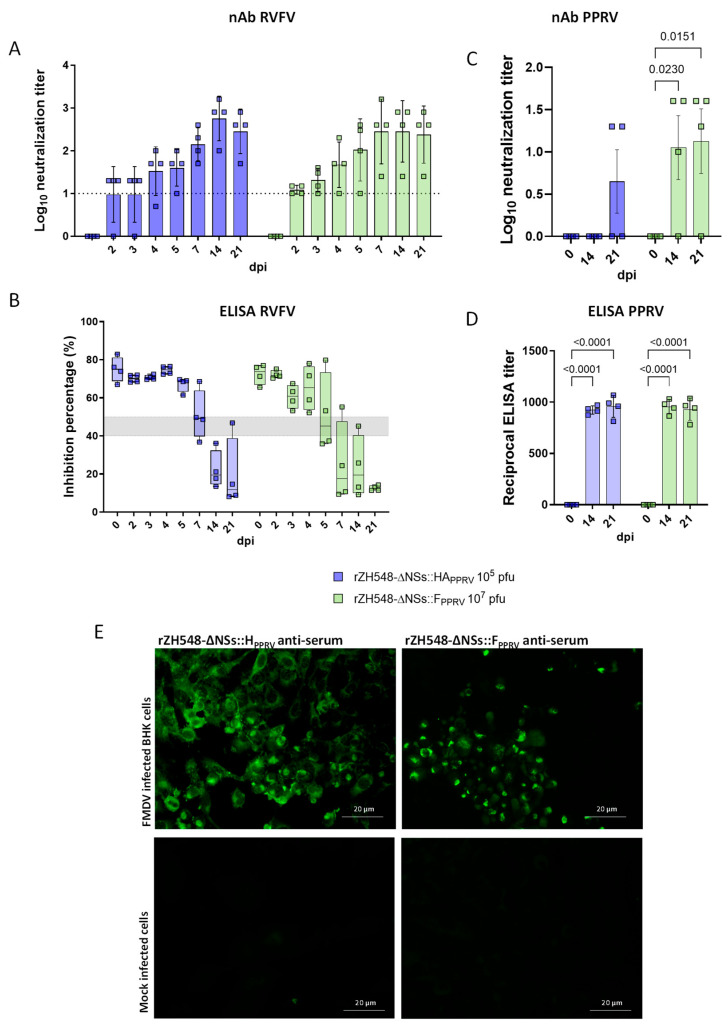
Humoral immune response following single-dose inoculation of sheep with rZH548-ΔNSs::H_PPRV_ or rZH548-ΔNSs::F_PPRV_. (**A**) Kinetics of RVFV-specific neutralizing antibody induction post-vaccination, showing titers at different time points. Horizontal dotted line represent assay sensitivity limit. (**B**) Competition ELISA detection of serum antibodies against RVFV nucleoprotein (N) at 2, 3, 4, 5, 7, 14, and 21 days post-inoculation. Shade area depicts the doubtful range, according to the manufacturer’s instructions (IDVet). (**C**) Determination of neutralizing antibody titers against PPRV Nigeria 75/1 for the same groups described in panel A, expressed as the log_10_ of the highest serum dilution that inhibited 100% of virus-specific cytopathic effect in 96-well flat-bottom plates. (**D**) Evaluation of PPRV-specific IgG levels in serum collected at days 14 and 21 by ELISA using Nigeria 75/1 PPRV-coated plates. Results are expressed as 1/X IgG titer for each individual animal. (**E**) Detection of FMDV-infected BHK-21 cells by indirect immunofluorescence with serum from a sheep inoculated with rZH548-ΔNSs::H_PPRV_ or rZH548-ΔNSs::F_PPRV_. Data (**A**,**C**,**D**) are expressed as mean ± SEM, with individual values shown as dots. Data (B) is expressed as median with interquartile range, with individual data points representing biological replicates. Significance (**C**,**D**) was analyzed using two-way ANOVA.

**Figure 8 vaccines-13-01039-f008:**
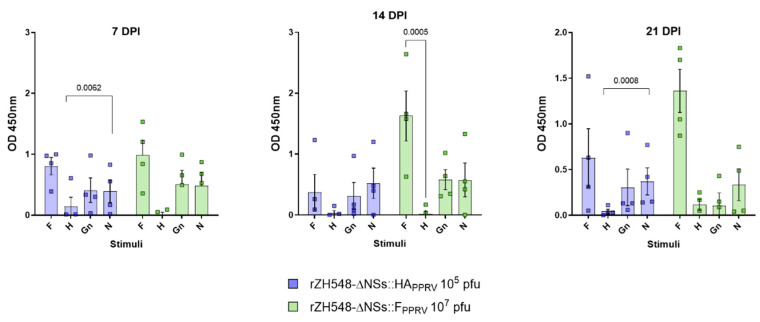
Detection of plasmatic levels of IFNγ after immunization. Blood samples were collected and re-stimulated with RPMI, ConA, or recombinant PPRV (F and H) or RVFV (Gn and N) proteins, and the levels of IFN-γ were measured in the plasma recovered from the samples. Background (i.e., the value determined in unstimulated controls) was subtracted from each sample. Data are presented as mean ± standard error of the mean (SEM). Statistical significance between groups across conditions was assessed using two-way ANOVA, followed by Tukey’s multiple comparisons test to correct for multiple testing. Individual data points are overlaid on the bars.

**Table 1 vaccines-13-01039-t001:** PPRV peptides used in the ICS assay for re-stimulation of mouse spleen cells, ^b^: the haplotype of the mouse major histocompatibility complex (MHC).

Peptide Name	PPRV Protein (aa Position)	Sequence	Predicted H-2 Allele Binding
F8	F (117–131)	VALGVATAAQITAGV	I-A^b^
F9	F (341–355)	QNALYPMSPLLQECF	I-A^b^
F10	F (284–298)	LSEIKGVIVHKIEAI	I-A^b^
H5	H (551–559)	YFYPVRLNF	D^b^
			K^b^
H9	H (427–441)	ITSVFGPLIPHLSGM	I-A^b^

## Data Availability

All relevant data are within the manuscript.

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
