# Peer review of "A Single Dose of Live-Attenuated Rift Valley Fever Virus Vector Expressing Peste Des Petits Ruminants Virus (PPRV) H or F Antigens Induces Immunity in Sheep"

_vaccines, 2025, doi:10.3390/vaccines13101039_

Round 1
Reviewer 1 Report (Previous Reviewer 1)
Comments and Suggestions for Authors
In the current resubmission, the authors have adressed the two major comments I proposed previously. I am satisfied with the the responses from the authors and also happy to see the improvments of other aspects of the current version. I have no further comments.
Author Response
We sincerely thank the reviewer for the thoughtful feedback. We greatly appreciate the time and effort invested in reviewing our work and are pleased that the revisions have addressed the major concerns.
Reviewer 2 Report (Previous Reviewer 3)
Comments and Suggestions for Authors
NA
Author Response
We thank the reviewer for their assessment. As no specific issues were indicated, we have thoroughly reviewed the entire manuscript to identify and address any potential errors or areas for improvement.
Reviewer 3 Report (New Reviewer)
Comments and Suggestions for Authors
The authors reported the peste des petits ruminants virus (PPRV) vaccine expressing H or F antigens, which induced protective immunity in sheep upon a single immunization. Upon a careful reading, this reviewer has the following comments, particularly within the methodology:
Lines 201-202: "lysed in SDS-PAGE sample buffer". I am not able to understand how the cells could be lysed in the SDS-PAGE buffer. Please provide clarity on the buffer composition and strength.
Line 203: "Proteins were transferred to nitrocellulose membranes". Please provide more details, such as voltage used, duration of transfer, buffer composition used for protein transfer, etc. Also, please provide the catalogue number of the reagents used, here and at other places.
Line 205: "Upon blocking, an anti-V5 epitope specific mAb (MCA1360, Bio-Rad)". This sentence is left incomplete. Please complete this sentence with information on the antibody dilution, incubation, etc.
Lines 205-207: The membrane was blocked with 5% milk solution prepared in TBS buffer. The information on the preparation of the primary antibody is missing. The secondary antibody was diluted in TBS-T. Why was the nitrocellulose membrane not blocked with the same buffer, such as TBS-T, as was used for the secondary antibody preparation, while no information on the preparation of the primary antibody is provided? How would this change impact the detection of protein expression level? Any thoughts? This section of the methodology is not properly written, and information is omitted.
Line 206: Which secondary antibody was used? It is not detailed.
Line 208: "Membranes were washed three times". Please mention the duration of washing and the rpm used.
Lines 208-209: This reviewer does not understand the methodology used here. What is this incubation step after the secondary antibody was already used in the previous step? Confusing and misleading.
Lines 210-212: The details of the substrates used are missing. Please clearly provide the catalogue numbers of all the reagents used in the experiments, along with other details.
Lines 224-230: More information on the methodology is required.
English can be improved in many places in the manuscript text. There are some typographical errors that need to be corrected with careful proofreading.
Comments on the Quality of English LanguagePlease carefully proofread the manuscript to correct the typographical errors.
Author Response
We thank the reviewer for their thorough and constructive feedback. Below we provide detailed responses to each point raised, along with corresponding revisions made to the manuscript.
Lines 201–202: "lysed in SDS-PAGE sample buffer"
We agree that the phrasing was unclear. The cells were lysed directly in commercial SDS-PAGE sample buffer. We have specified the buffer catalog number and its use in the revised manuscript.
Line 203: "Proteins were transferred to nitrocellulose membranes"
We have now included the missing details: proteins were transferred using wet electroblotting at 100 V for 1 hour at room temperature. The transfer buffer consisted of 25 mM Tris, 192 mM glycine, and 20% methanol, prepared in ultrapure water. The nitrocellulose membranes used were from Amersham. These details have been added to the Materials and Methods section.
Line 205: "Upon blocking, an anti-V5 epitope specific mAb (MCA1360, Bio-Rad)"
Thank you for pointing this out. The sentence has been completed to include the following: “Upon blocking, an anti-V5 epitope specific mAb (MCA1360, Bio-Rad) was used diluted 1:1000 in TBS-T.”
Lines 205–207: Blocking buffer and antibody preparation
We thank the reviewer for pointing out this inconsistency. The omission regarding the buffer composition was due to a writing oversight. In fact, both the blocking and washing steps were performed using TBS-T (TBS supplemented with 0.05% Tween-20). The membranes were blocked with 5% skim milk in TBS-T to minimize non-specific binding, and all antibody incubations and washes were carried out in the same buffer. We have corrected the text to reflect this accurately and ensure consistency throughout the methodology.
Line 206: Secondary antibody details
We have now specified the secondary antibody used: anti-mouse polyvalent immunoglobulin conjugated to horseradish peroxidase (HRPO; Sigma-Aldrich), diluted 1:1000 in TBS-T and incubated for 1 hour at room temperature.
Line 208: Washing conditions
The membranes were washed three times with TBS-T, each wash lasting 30 minutes at 40 rpm. These details have been added to the revised manuscript.
Lines 208–209: Incubation step clarification
We acknowledge the confusion caused by the original phrasing. The incubation step mentioned refers to the secondary antibody incubation, which follows the primary antibody step. We have restructured the paragraph to clearly indicate the sequence of blocking, primary antibody incubation, washing, secondary antibody incubation, and final washes.
Lines 210–212: Substrate details
We have now included the full details of the chemiluminescent substrate used: ECL Plus Western Blotting Detection Reagent (GE Healthcare, Cat# 28980926). Signal detection was performed using a ChemiDoc Imaging System (Bio-Rad). This catalogue number and product details have been added for transparency and reproducibility.
Lines 224–230: Methodology clarification
We have expanded this section to include the number of passages analyzed, the MOI used for infection (MOI = 1), the time point for cell harvest (24 h post-infection), and the antibodies used for detection. We also clarified that the 2B1 anti-RVFV-N monoclonal antibody was used as an internal control for infection normalization. All relevant details have been added.
Language and typographical corrections
We have carefully proofread the entire manuscript and corrected typographical errors, improved sentence structure, and ensured consistency in scientific terminology and formatting. We appreciate the reviewer’s suggestion and have made substantial improvements to the clarity and flow of the text.
We hope these revisions address all concerns satisfactorily and improve the overall quality and transparency of the manuscript. We thank the reviewer again for their valuable input.
Round 2
Reviewer 3 Report (New Reviewer)
Comments and Suggestions for Authors
No further comments.
This manuscript is a resubmission of an earlier submission. The following is a list of the peer review reports and author responses from that submission.
Round 1
Reviewer 1 Report
Comments and Suggestions for Authors
The current manuscript reports the design and test of a bivalent vaccine strategy using a live-attenuated Rift Valley fever virus (RVFV) backbone expressing the H or F proteins of peste des petits ruminants virus (PPRV). These authors test the stability, immunogenicity, and safety of the recombinant RVFV through in vitro and in vivo experiments. Although without virus challenge data, the current setups provide valuable data regarding the conceptulization of multivalent vaccine strategy using the live-attenuated RVFV backbone. I only have two major comments:
1) The H antigen expression is very low in passage 4 and almost disappear in passage 5, questioning the stability of the rRVFV. Thus the conclusion that the rRVFV expression is stable over 5 passage is not convincing. The autors need to explain the reason or at least provide sequencing data to confirm if loss of expression is due to gene deletion, mutations, or reduced transcription/translation.
2) The F protein induces stronger CD8⁺ T-cell responses than H protein. The authors need to explain this difference and make sure this is not due to technical issues.
Author Response
- The H antigen expression is very low in passage 4 and almost disappear in passage 5, questioning the stability of the rRVFV. Thus the conclusion that the rRVFV expression is stable over 5 passage is not convincing. The autors need to explain the reason or at least provide sequencing data to confirm if loss of expression is due to gene deletion, mutations, or reduced transcription/translation.
Thank you for raising this important concern regarding the stability of H antigen expression across serial passages.
We acknowledge that the expression of the H antigen noticeably declines by passage 4 and becomes virtually undetectable at passage 5, as shown in our Western blot analyses. Our original statement (line 344-347) that recombinant viruses expressing heterologous antigens could be maintained at least during four consecutive passages was intended to reflect this observed threshold. We agree that describing the recombinant virus as "stable over 5 passages" in the abstract was a mistake, and is now changed.
In the meantime, it's worth noting that sufficient antigen expression was consistently observed up to passage 4, which enabled us to produce adequate stocks for immunization trials with no detectable compromise in immune induction. We believe that the reduced expression observed during serial passages may be due to partial deletion of the inserted gene.
Furthermore, we are currently working on a follow-up study that explores the expression of the H and F antigens in more detail. This includes investigating their stability, processing, and how they behave in host cells. The results will help refine our vaccine design and will be shared in a future publication.
2) The F protein induces stronger CD8⁺ T-cell responses than H protein. The authors need to explain this difference and make sure this is not due to technical issues.
We appreciate the reviewer’s insight regarding the stronger CD8⁺ T-cell responses elicited by the F protein compared to the H protein and have examined this difference closely and included this in the discussion. The observed disparity is biologically plausible given the roles of each antigen: the fusion (F) protein is involved in viral entry and membrane fusion, processes that naturally favor presentation via MHC class I pathways and stimulate cytotoxic T-cell responses. In contrast, the hemagglutinin (H) protein primarily facilitates receptor binding and is more associated with humoral immunity. We also believe that the higher expression levels of the F protein may influence the elicited T-cell responses.
Expression analyses confirmed that both proteins were stably produced in infected cells; notably, the F protein underwent expected post-translational processing (F0 to F1), which may enhance antigen presentation and cellular immunogenicity. All ICS assays were conducted under identical experimental conditions, with consistent cell counts and staining protocols, ruling out technical variance as a contributing factor.
Reviewer 2 Report
Comments and Suggestions for Authors
1. Abstract
The novelty is implied but not explicitly stated — it doesn’t contrast this vaccine with existing ones.
No numerical results or performance metrics (e.g., neutralizing antibody titers, protection rates) are given.
2. Introduction
The DIVA concept is underemphasized despite being a key practical advantage.
Previous attempts at recombinant bivalent vaccines are mentioned (e.g., vaccinia, capripox), but direct comparisons to this approach (live-attenuated RVFV vector) are missing.
Scientific justification for selecting H or F genes from PPRV is not clearly developed.
3. Materials and Methods
Lack of statistical power analysis or sample size justification.
4. Results
Overreliance on phrasing like “significant” without reporting actual p-values in text.
One challenge group (rMP12-F) appears to have less robust data than the H group, but the difference is not deeply discussed.
5. Discussion
Under-discusses limitations, e.g.:
No long-term immunity data (>28 days).
No efficacy in actual PPR or RVF challenge model (uses surrogates).
Unclear scalability or manufacturability of recombinant constructs.
DIVA validation is shown only via serology (anti-N detection), without field validation.
Author Response
- Abstract
Reviewer Comment: The novelty is implied but not explicitly stated — it doesn’t contrast this vaccine with existing ones.
Thank you for the comment. We have revised the abstract to clearly emphasize the novelty of our approach, stating that the use of a live-attenuated Rift Valley Fever Virus (RVFV) vector expressing PPRV antigens offers a new multivalent vaccination strategy. This approach contrasts with prior platforms like adenovirus, poxvirus, and herpesvirus vectors by combining safety and dual disease coverage in a single dose.
Reviewer Comment: No numerical results or performance metrics (e.g., neutralizing antibody titers, protection rates) are given.
We have revised the abstract to include representative quantitative metrics, such as neutralization titers.
- Introduction
Reviewer Comment: The DIVA concept is underemphasized despite being a key practical advantage.
We agree and have reinforced the importance of the DIVA strategy in the revised introduction.
Reviewer Comment: Previous attempts at recombinant bivalent vaccines are mentioned, but direct comparisons to this approach are missing.
Thank you for this remark. We have now included two additional references concerning other bivalent and multivalent vector vaccines. However, it should be noted that direct comparisons are challenging, as relatively few vaccines have been developed using this strategy, and those that exist are based on diverse vector platforms.
Reviewer Comment: Scientific justification for selecting H or F genes from PPRV is not clearly developed.
We have clarified this by stating that both H and F proteins are known protective antigens. H is the primary target for neutralizing antibodies, while F induces cytotoxic T cell responses. Evaluating both separately allowed us to assess and compare their individual immunogenicity profiles within the RVFV platform.
- Materials and Methods
Reviewer Comment: Lack of statistical power analysis or sample size justification.
This study was designed as a pilot proof-of-concept rather than a confirmatory trial. Accordingly, no formal sample size calculation was performed. The group sizes were chosen based on prior experience with similar vaccine models and represent the minimum number of animals required to evaluate immunogenicity trends while adhering to ethical principles of animal use. In future efficacy studies, formal power analysis and sample size justification will be conducted to ensure appropriate statistical power for protection outcome comparisons.
Although the sample sizes used in this study are relatively limited, the article by Richardson and Overbaugh ( DOI: 10.1128/JVI.79.2.669-676.2005 )supports the notion that the number of mice employed here is sufficient to achieve meaningful statistical power. Regarding the sheep experiments, we acknowledge that the group sizes are modest; however, this is justified by ethical considerations and the logistical challenges associated with animal handling, housing, and long-term maintenance in a biosafety-compliant setting.
- Results
Reviewer Comment: Overreliance on phrasing like “significant” without reporting actual p-values.
Thank you for highlighting this. We have revised the manuscript and we believe this might have been a misunderstanding. Throughout the text we remark results that are not significant and regarding the p-values, they are displayed in the figures.
Reviewer Comment: One challenge group (rMP12-F) appears to have less robust data than the H group, but the difference is not deeply discussed.
Thank you for this observation. In our view, the interpretation may in fact be the opposite. Throughout the manuscript, we emphasize that the virus expressing the Fusion (F) protein consistently showed more robust data, including greater stability across serial passages, higher expression levels and more pronounced cellular immune responses. These findings are mentioned in the figures and discussion sections, and support the superior immunogenic performance of the F construct compared to the one expressing Hemagglutinin (H). Finally, it should be noted that the viruses used in this study are based on the rZH548 backbone, not rMP12.
5. Discussion
Reviewer Comment: Under-discusses limitations, including:
We have revised the discussion to address the limitations mentioned by the
reviewer and included all the relevant information he required.
- No long-term immunity data (>28 days).
As he mentions the study only evaluates short-term immune responses since
this was just a first approach and longer-term durability remains to be
assessed in follow-up studies.
- No efficacy in actual PPR or RVF challenge model.
Protective efficacy against live virus challenge was not assessed. These
studies are planned using higher doses and optimized protocols
- Unclear scalability/manufacturability of recombinant constructs.
While current viral titers were sufficient for experimental use, large-scale
production may require optimization in alternative culture conditions,
specially for the rZH548-ΔNSs::HPPRV since it reaches lower titers.
- DIVA validation limited to serology, no field testing.
DIVA potential wasn’t assessed. But the design of the recombinant RVFV
vectors, which do not express the PPRV nucleoprotein (N), supports DIVA
functionality in principle.
Reviewer 3 Report
Comments and Suggestions for Authors
Please find the comments in the attachment.

Author Response
Major Comments
1) The abstract lacks quantitative data on immune responses (e.g., neutralizing antibody titers or IFNγ levels) and a clear concluding statement. The authors are recommended to include numerical results to strengthen the scientific impact of the abstract and add a more definitive conclusion that highlights which construct (H or F) elicited a stronger immunological response.
We appreciate the reviewer’s suggestion. In response, we have revised the abstract to include quantitative data on the cellular immune response. Furthermore, we have clarified that the construct expressing the F protein elicited a stronger immunological response, which is now emphasized in the concluding statement of the abstract.
2) Using the same dose for each recombinant vector is important to enable a proper comparison; however, the authors used different doses of rZH548ΔNSs:FPPRV and rZH548ΔNSs:HPPRV for animal immunization. The authors should comment on the rationale for this difference.
We appreciate the reviewer’s observation. The difference in doses used for rZH548ΔNSs:FPPRV and rZH548ΔNSs:HPPRV was due to the viral titers obtained during the preparation of the recombinant vectors, as noted in line 465. Although we could have reduced the dose of rZH548ΔNSs:FPPRV to match that of rZH548ΔNSs:HPPRV, our primary aim in this initial study was to determine whether expressing PPRV antigens using the RVFV vector could elicit an immune response, rather than to directly compare the immunogenicity of the F and H proteins. Future work will address these matters more in depth, including inoculation doses, co-immunizing with both constructs, using a more relevant animal model such as goats, and performing challenge studies. These steps will allow for a more comprehensive evaluation of the strategy, which we consider beyond the scope of the present study.
3) Including appropriate controls is critical for scientifically validating the findings. Although the authors performed multiple experiments, it is unclear whether controls were used. It is highly recommended that the authors clarify the use of controls and explain why controls were omitted in certain experiments, if applicable.
We appreciate the reviewer’s concern regarding the use of appropriate controls. In our experiments, we primarily employed monoclonal antibodies specific for transgenes, which do not cross-react with RVFV antigens or cellular components. For this reason, negative controls were not included, as the risk of non-specific detection was minimal and well characterized.
Regarding positive controls, while transfected plasmids and PPRV virus were considered, the plasmids showed limited functionality, and PPRV does not replicate efficiently in Vero cells, rendering these controls suboptimal. Notably, in the SD6 experimental setup, which posed particular challenges, we included both mock-infected cells and cells infected with FMDV as appropriate references. We hope this clarifies the reasons behind our control strategy.
4) The rationale for selecting the H and F antigens individually rather than co-expressing both or using a fusion construct is not clearly explained. The authors are recommended to clarify whether a co-expression or multivalent strategy was considered, and to justify why it was not pursued.
We appreciate the reviewer’s comment. As previously mentioned, this study was intended as an initial approach to assess whether PPRV antigens could be effectively expressed using an RVFV-based vector, whether they would elicit an immune response, and whether that response could be linked to protection. At this stage, we did not consider co-expression of H and F or a fusion construct. Our primary objective was to evaluate each antigen individually to identify whether the approach was viable. Future studies are planned to assess whether simultaneous immunization using both H and F recombinant viruses might enhance immune responses, as suggested in our discussion. This strategy is particularly appealing given the complementary contributions of each antigen observed in the present work.
5) The authors reported poor stability of H and F expression after 4–5 passages, which raises concerns about vaccine production scalability and quality control. The authors should clarify whether codon optimization was considered; if not, it is recommended that they explore this approach. Moreover, alternative insertion sites or regulatory sequences should be investigated to improve the stability of target protein expression.
Thank you for this thoughtful comment. We acknowledge that the recombinant virus shows reduced stability of H and F expression after several passages. However, it is feasible to generate high-titer viral stocks within three to four passages. Additionally, we are currently optimizing virus concentration protocols that increase viral titers without compromising the expression of heterologous antigens, thereby enhancing scalability. We fully agree that optimizations such as codon optimization, evaluation of alternative insertion sites, or improved regulatory sequences are valuable strategies to enhance expression stability. We will address these considerations in future studies. It is worth noting, that the current work represents a very preliminary stage of vaccine development. While scalable production is an important future consideration, it is too early to evaluate at this point.
6) In Figure 2B, multiple bands are observed for both H-and F-PPRV during protein detection by WB. The authors are recommended to include appropriate positive and negative controls, such as Mock transfected cells and pcDNA_H-PPRV or pcDNA_F-PPRV plasmids, to determine whether these additional bands are specific or nonspecific.
We thank the reviewer for their suggestion. As all transgenes were detected using monoclonal antibodies, V5tag and SD6, we believe the need for additional negative controls is less critical in this context. In fact, infection of cells with RVFV alone yields no signal with the V5 or SD6 monoclonal antibodies, supporting the specificity of detection. The bands observed in Figure 2B are reproducibly recognized by the corresponding antibodies and differ between H and F, suggesting they are products of transgene expression rather than nonspecific signals. Furthermore, comparing transfected plasmid expression with recombinant virus infection is not straightforward, as protein expression profiles between these two systems are inherently different and not directly comparable.
Although full Western blot membranes were not uploaded in the figures, they have been provided to the journal as supplementary files. These membranes include both negative and positive controls, further confirming the specificity of the antibodies, since each virus shows distinct bands corresponding to its respective transgene.
7) The results of the neutralization assay indicate that some animals, particularly in the H-PPRV group, failed to induce neutralizing antibody response. This raises concerns regarding vaccine efficacy. The authors are therefore encouraged to include a discussion addressing responder Vs non-responder animals and to explore potential contributing factors (e.g., dosage, vector expression, or host-specific variables).
Thank you for this important observation. As noted in the discussion (lines 592 and 599), the lower induction of neutralizing antibodies in the H-PPRV group may be attributed to several factors, including the lower dose administered and the expression profile of the H protein, which likely influences viral replication kinetics. These aspects could directly affect the immunogenic potential of the RVFV vector.
Moreover, we would like to point out the importance of inter-individual variability, particularly in sheep, which are outbred animals. Such genetic diversity can result in heterogeneous immune responses, further contributing to the presence of responder and non-responder individuals within experimental groups.
8) No viral challenge was performed in target animals in this study. Although this limitation is acknowledged, the current data remain incomplete. It should be clearly emphasized as a limitation in both the abstract and discussion, along with a justification or plan for future challenge studies.
The absence of viral challenge in target animals is acknowledged as a limitation of the current study. This decision was based on the need to first evaluate the safety and immunogenicity profile of the recombinant vectors. Nonetheless, the promising results obtained in terms of immune response provide a strong rationale for conducting challenge experiments in future studies. This limitation and the future experiments have been explicitly addressed both in the discussion section and in the abstract.
9) The statistical methods are stated broadly in the manuscript. The authors are recommended to specify which statistical test was used for each figure and to include correction methods for multiple comparisons where appropriate
Statistical tests have now been clearly specified in each figure legend to improve transparency. Where multiple group comparisons were made, the appropriate test type is indicated. In instances where correction methods were not applied, this was due to the limited number of comparisons or the exploratory scope of the analysis, as detailed in the Methods section to avoid any potential misinterpretation.
Minor Comments:
- For clarity and specificity, the title of the manuscript should be revised for improved conciseness. The following is a suggested alternative: “A single dose live attenuated Rift Valley Fever Virus vector expressing peste des petits ruminants virus (PPRV) H or F antigens induces immunity in sheep.”
We agree with the reviewer’s suggestion regarding the manuscript title. The proposed revision is clearer and more concise. We have modified the title accordingly to: “A single dose live attenuated Rift Valley Fever Virus vector expressing peste des petits ruminants virus (PPRV) H or F antigens induces immunity in sheep.”
- The use of FMDV site A as a tag is innovative, but the lack of neutralization raises concerns. Discuss whether it might interfere with antigen folding or immunodominance of RVFV/PPRV epitopes.
We appreciate the reviewer’s insightful comment. Although FMDV site A is a neutralizing epitope, our data show that when fused to PPRV H or F proteins, anti-FMDV antibodies were generated but lacked neutralizing activity against the homologous virus. If it had conferred neutralizing capacity, it would have conferred protective immunity against FMDV, making the construct potentially multivalent against three major ruminant pathogens and it would’ve been a great advantage.
However, in the context of this Therefore, the lack of neutralizing antibodies is not a concern but rather an opportunity to better understand this limitation and explore strategies to optimize the prototype.
As discussed in the manuscript, one possible explanation lies in the conformational constraints imposed by fusion, which may prevent the epitope from adopting the native structure it displays in VP1 of FMDV. Regarding immunodominance, we cannot exclude the possibility that fusion to PPRV H or F may compromise the presentation of the FMDV epitope. This is an interesting topic for future studies, and while we acknowledge its relevance, a detailed analysis falls outside the scope of this manuscript.
- Several typos and grammatical inconsistencies are present. For example: “food were supply ad libitum; line 95” → “food was supplied ad libitum”. “immunogenicity studies were conducted...to evaluate the induction cellular and humoral immune responses; line 19-20” → should be rephrased for clarity. Please review the entire manuscript.
We sincerely thank the reviewer for pointing out these typographical and grammatical issues. We have carefully revised the manuscript to address all identified inconsistencies and improve clarity throughout. Specifically:
The sentence in line 95 has been corrected to: “Water and food were supplied ad libitum.”
The wording in lines 19–20 has been rephrased to more clearly convey the purpose of the immunogenicity studies.
In addition, we have conducted a comprehensive review of the entire manuscript to ensure accuracy, consistency, and readability.
- The sequence length (in bp or aa) for H and F antigens should be provided.
We thank the reviewer for pointing out this information was missing and it has been added in the materials and methods section.
- To avoid confusion between name and number of mice used in the study, please revise “129 Sv/Ev mice” →“129/SvEv mice”
It has been changed throughout the manuscript for clarity.
- Please clarify the source of all of cell lines used in this study.
It has been added in materials and methods
- Line 103, both sequences were chemically synthetized, please clarify whether the sequences were synthesized by the authors or commercially obtained.
In the previous line, it is specified that a commercial supplier, Biomatik, provided the plasmids. However, we have revised the text for greater clarity to explicitly state that the sequences were commercially synthesized.
- Line 123: Please clarify why a co-culture of BHK-21 and HEK293T cells was used for transfection.
HEK293T cells are needed for the transfection of the pol I-driven genomic plasmids and pol II-driven support plasmids. To enhance amplification of the recombinant virus generated by the HEK293T cells, they were co-cultured with BHK-21 cells, which lack the ability to express viral segments from the human pol I promoter.
- Line 146, please clarify which monoclonal antibody was used.
We sincerely apologize for any confusion, as we were unable to identify the specific reference cited by the reviewer for line 146. We have assumed it references to the indirect immunofluorescence assay described in the text, in which a monoclonal antibody is used, specifically, the V5 epitope tag antibody (clone MCA 1360, Bio-Rad).
If instead the reference concerns the monoclonal antibody employed to detect the viral nucleoprotein in the immunostaining, this is also specified in the manuscript as the 2B1 anti-RVFV-N monoclonal antibody.
- The discussion of the findings presented in Figure 2D is missing from the manuscript text.
Thank you very much for your observation. As shown in Figure 2D (as in 2C), the recombinant virus expressing the H protein forms smaller plaques, which is consistent with its lower replication titers. This change in plaque morphology and growth kinetics may be related to an interference caused by the expression of the H protein with efficient viral replication.
We would like to point out that this phenomenon was mention in the results and discussion in relation with Figure 2C, which illustrates the same comparison via immunostaining. We have now included a reference to Figure 2D in the corresponding sections of the text to ensure a more complete and coherent interpretation of these findings.
Round 2
Reviewer 2 Report
Comments and Suggestions for Authors
You've done a good job of correcting most of the points I've pointed out. Thanks for the effort.
Author Response
Thank you for your valuable comments and your positive response.
Reviewer 3 Report
Comments and Suggestions for Authors
Thank you to the authors for addressing my comments in a constructive manner. However, to improve the quality of the work, it is essential to experimentally include appropriate controls for each experiment presented in the study. Additionally, the authors should work to enhance the stability of the recombinant viruses generated, rather than deferring this to future studies. This is a critical issue without resolving it now, the applicability and relevance of this work in future vaccine development will remain limited.
Author Response
Comment: Thank you to the authors for addressing my comments in a constructive manner. However, to improve the quality of the work, it is essential to experimentally include appropriate controls for each experiment presented in the study. Additionally, the authors should work to enhance the stability of the recombinant viruses generated, rather than deferring this to future studies. This is a critical issue without resolving it now, the applicability and relevance of this work in future vaccine development will remain limited.
Answer: We would like to thank the reviewer for their thoughtful and constructive feedback. While we agree that more controls could always be added, we have revised each experiment and believe that we have included all the necessary controls to support our conclusions.
We also appreciate the reviewer’s comments on the importance of recombinant virus stability. In our study, the viruses were evaluated for a limited number of passages — sufficient to support the immediate experimental objectives.
While a higher number of stable passages would indeed improve applicability for large-scale vaccine preparation, these experiments are not necessary for this initial approach. As we previously mentioned, three passages are sufficient to prepare a working stock, and importantly, the recombinant viruses maintained consistent phenotypic and genotypic profiles until passage four, as detailed in the Results section. These findings suggest that the level of stability is adequate for the conclusions reached.